# Metabolic Complete Response of Metastatic Oncogene-Negative, PDL1-Negative Non-Small Cell Lung Cancer After Chemo-Immunotherapy and Radiotherapy: A Case Report

Alessia Surgo [1,2,*], Valerio Davì [1], Maria Paola Ciliberti [1], Roberta Carbonara [1], Morena Caliandro [1], Fiorella Cristina Di Guglielmo [1], Nicola Sasso [3], Roberto Calbi [4], Maria Annunziata Gentile [4], Tiziana Talienti [1], Isabella Bruno [5], Michele Troia [6], Ilaria Bonaparte [1], Giuseppe Mario Ludovico [7], Giammarco Surico [3] and Alba Fiorentino [1,2]

1   Radiation Oncology Department, General Regional Hospital F. Miulli, 70021 Acquaviva delle Fonti, BA, Italy; v.davi@miulli.it (V.D.); mp.ciliberti@miulli.it (M.P.C.); roberta.carbonara@miulli.it (R.C.); m.caliandro@miulli.it (M.C.); f.diguglielmo@miulli.it (F.C.D.G.); t.talienti@miulli.it (T.T.); i.bonaparte@miulli.it (I.B.); a.fiorentino@miulli.it (A.F.)
2   Department of Medicine and Surgery, LUM University, 70010 Casamassima, BA, Italy
3   Medical Oncology Department, General Regional Hospital F. Miulli, 70021 Acquaviva delle Fonti, BA, Italy; n.sasso@miulli.it (N.S.); g.surico@miulli.it (G.S.)
4   Radiology Department, General Regional Hospital F. Miulli, 70021 Acquaviva delle Fonti, BA, Italy; r.calbi@miulli.it (R.C.); m.gentile@miulli.it (M.A.G.)
5   Nuclear Medicine Department, General Regional Hospital F. Miulli, 70021 Acquaviva delle Fonti, BA, Italy; i.bruno@miulli.it
6   Pathology Department, General Regional Hospital F. Miulli, 70021 Acquaviva delle Fonti, BA, Italy; m.troia@miulli.it
7   Urology Department, General Regional Hospital F. Miulli, 70021 Acquaviva delle Fonti, BA, Italy; g.ludovico@miulli.it
*   Correspondence: a.surgo@miulli.it; Tel.: +39-0803054608

**Abstract:** A 71-year-old male ex-smoker presented in October 2021 to our department with a brain and bone metastatic adenocarcinoma NSCLC. PDL1, ROS, EGFR, and ALK were negative. He underwent stereotactic radiotherapy for brain metastases. In November 2021, he started a chemotherapy (CHT) regimen with cisplatin (75 mg/m$^2$ every 21 days) and pemetrexed (500 mg/m$^2$ every 21 days), and ICI with Atezolizumab (1200 mg every 21 days). In July 2022, RT to the lung tumor and mediastinal nodal was performed with a total dose of 45 Gy in 15 fractions. He continued with immunotherapy until December 2022, when a grade 3–4 toxicity from immunotherapy was observed (hypothyroidism, psoriasis, and cystitis). He achieved a complete clinical response to the therapy. To date, the patient is alive, with a complete metabolic response, without treatment at 37 months from diagnosis.

**Keywords:** NSCLC; brain metastases; immunotherapy; complete response; case report

## 1. Introduction

Lung cancer (LC) is the second most common neoplastic disease worldwide, accounting for 18% of all cancer deaths [1]. Non-small cell lung cancer (NSCLC) is often diagnosed at advanced stages and presents significant treatment challenges due to its complexity and resistance to traditional therapies [2–4]. While the emergence of targeted therapy and immunotherapy has transformed the treatment of NSCLC, resulting in significant enhancements in survival rates, patients with stage IV oncogene-negative, PDL1-negative NSCLC continue to experience a grim prognosis [4]. The role of ablation treatment, including surgery and radiotherapy (RT), has been assessed in the current scenario. Numerous studies indicate that the 5-year overall survival (OS) rate for patients with limited metastases who underwent surgery or ablation at all disease sites ranges from 29% to 45% [5,6].

Consequently, treatment strategies for these patients should be performed, and the use of potentially curative treatment, rather than merely palliative, should be increased.

The present case report delves into such a scenario, demonstrating a durable complete metabolic response in a patient with brain and bone metastases from NSCLC who was administered immune checkpoint inhibition (ICI) with chemotherapy, stereotactic radiotherapy (SRT) for brain metastases, and hypofractionated RT (hypo-RT) for lung tumor.

## 2. Case Report

A 71-year-old ex-smoker male patient with a history of hypertension and bladder cancer (pT1a in 2016) presented at our hospital in October 2021 with headache, language and memory difficulties, and tinnitus. He presented with a nonproductive night cough without dyspnea.

Magnetic resonance (MRI) of the brain was performed, reporting the presence of three brain lesions (one in the left frontal lobe of about 4 cm; the other two in the cerebellum, right and left, of about 2 cm) with oedema (Figure 1a–c). Moreover, an occipital bone lesion was reported.

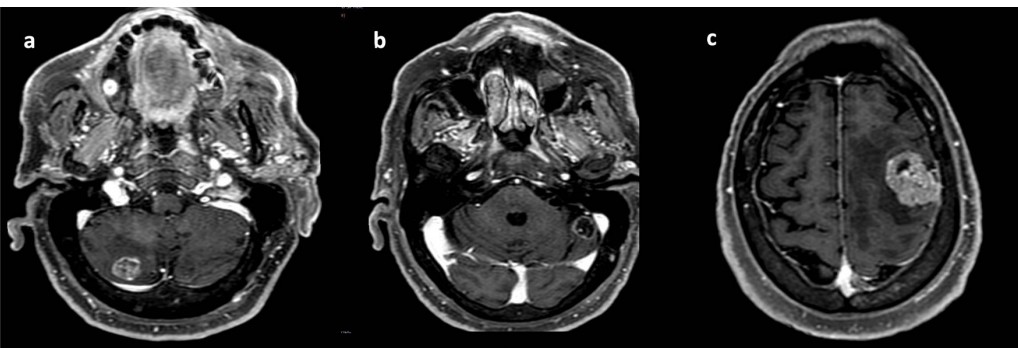

**Figure 1.** Brain magnetic resonance at diagnosis. Legend: (**a**) right cerebellar lesion; (**b**) left cerebellar lesion; (**c**) left frontal lobe lesion.

After one week, a total body computed tomography scan (TB CT scan) was performed, revealing a solid nodular, expansive mass (39 mm) in the right upper lobe of the lung and multiple lymphadenopathies in the mediastinum (paratracheal and subcarinal) (Figure 2a).

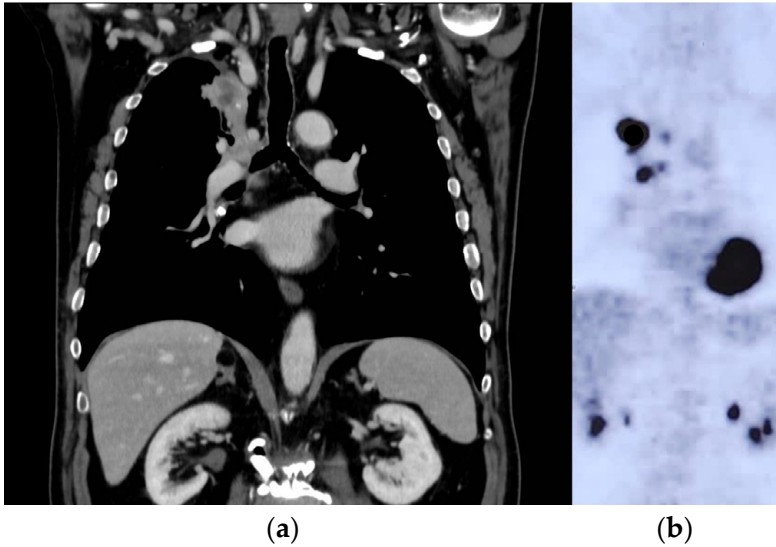

(**a**)      (**b**)

**Figure 2.** Lung tumor imaging at diagnosis. Legend: (**a**) lung tumor in the right upper lobe on CT scan; (**b**) PET-FDG hyper-capturing lesion on right superior lobe and mediastinal nodes.

Fluorodeoxyglucose positron emission tomography (FDG PET) (Figure 2b) confirmed the presence of multiple hyper-capturing lesions consistent with the CT scan findings: right superior lobe (SUV 16.8), mediastinal nodes, plus a lesion on the occipital bone. A cT2N2M1c stage IV lung cancer was diagnosed with CEA and Ca19.9 values of 15.4 ng/mL (<6) and 65.6 UI/mL (<37), respectively.

A CT-guided biopsy was performed on the lung mass, and a poorly differentiated neoplasm suggestive of NSCLC adenocarcinoma was diagnosed. PDL1, ROS, EGFR, and ALK were all negative.

EGFR genes and ALK/ROS1 mutations were detected using DNA sequencing techniques and the amplification–refractory mutation system. The mRNAs of ROS1 and ALK fusions were examined using polymerase chain reaction techniques and a fusion gene detection kit.

The patient then started LINAC-based SRT: the frontal right lesion received 36 Gy/6 fractions, and the cerebellum lesions received 24 Gy/3 fractions (Figure 3a,b) [6–8].

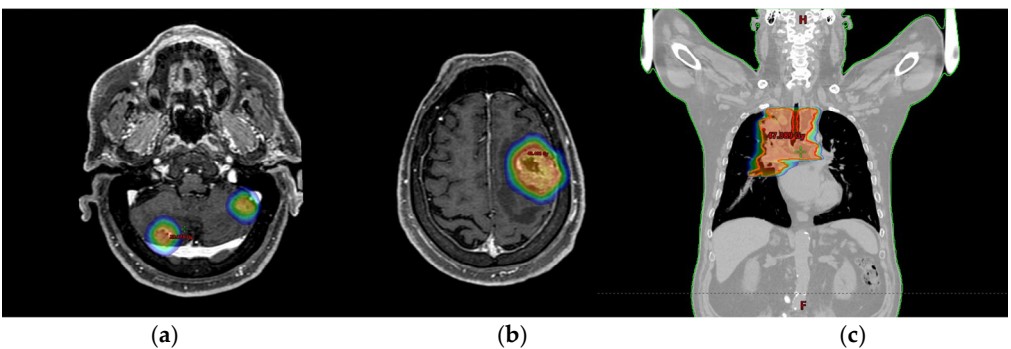

|   (**a**)   |   (**b**)   |   (**c**)   |

**Figure 3.** Radiotherapy treatment plans. Legend: (**a**) SRT treatment plan for cerebellar lesions; (**b**) SRT treatment plan for frontal right lesion; (**c**) mediastinal hypofractionated treatment plan. The colored areas show the isodoses which highlight the progressive decrease in the radiation dose moving from the center of the lesion to the periphery (from red to blue).

The gross tumor volume (GTV) was defined on contrast-enhanced T1-weighted sequence MRI and was assumed equal to the clinical target volume (CTV). The planning target volume (PTV) was obtained from the GTV plus an isotropic margin of 1 mm in all directions. The treatment was performed with a 6X Flattening Filter Free (FFF) beam and Volumetric Modulated Arc Therapy (VMAT) with noncoplanar arcs, optimized for True-Beam™ (Varian Medical System, Palo Alto, CA, USA). The approach of HyperArc™ was used for the simultaneous treatment of the two cerebellar lesions [7]. Prescription dose, normalization, and optimization were performed according to ICRU91 guidelines. During RT, IGRT with cone-beam CT (CBCT) and real-time surface-guided RT using AlignRT® (Vision RT, London, UK) were performed daily prior to and during the RT session [7].

In November 2021, the patient started a chemotherapy (CHT) regimen with cisplatin (75 mg/m$^2$ every 21 days) and pemetrexed (500 mg/m$^2$ every 21 days), and ICI with Atezolizumab (1200 mg every 21 days).

After four CHT and ICI cycles, blood tests showed a favorable response in cancer markers: CEA decreased from 15 ng/mL to 7 ng/mL, and Ca19.9 decreased from 65.5 UI/mL to 41.

Subsequent CT scans in January 2022 showed a positive response to brain metastases and a reduction in the size of the lung tumor from 39 mm to 29 mm, as well as improvements in thoracic lymphadenopathies, reporting a partial response (RECIST).

From February 2022, he continued with only Atezolizumab (10 cycles).

In June 2022, the imaging showed growth in the lung mass and mediastinal nodal size with progressive disease defined by RECIST. The brain MRI indicated a positive response to SRT, and no other brain lesions were found. The bone lesion was not irradiated but it was likely reduced probably due to the abscopal effect.

In July 2022, the patient received hypo-RT (45 Gy/16 fractions to lung mass and PET + nodes) (Figure 3c).

From 29 August 2022 to 2 February 2023, Atezolizumab treatment was performed with a total of 17 cycles.

ICI was discontinued, as agreed with the patient, considering his adverse reactions. The patient developed hypothyroidism (TSH 64 UI/mL, range 0.27–4.2) with heart failure treated with levothyroxine, grade 4 dermatitis from psoriasis, and pruritus unresponsive to antihistamines, which was managed with glucocorticoids. The patient resolved the side effects 3–4 months after Atezolizumab's interruption. Moreover, during months from March 2023 onward, he reported urinary symptoms, including mild hematuria and cystitis. In July 2024, the patient reported chronic cystitis with G3 hematuria and underwent a cystoscopy, and a biopsy was performed.

The pathological evaluation showed bladder fragments with massive ulcer-necrotic phenomena and chronic granulomatous, giant cell, and histiocytic inflammatory processes (CKAE1/AE3 negative, Figure 4).

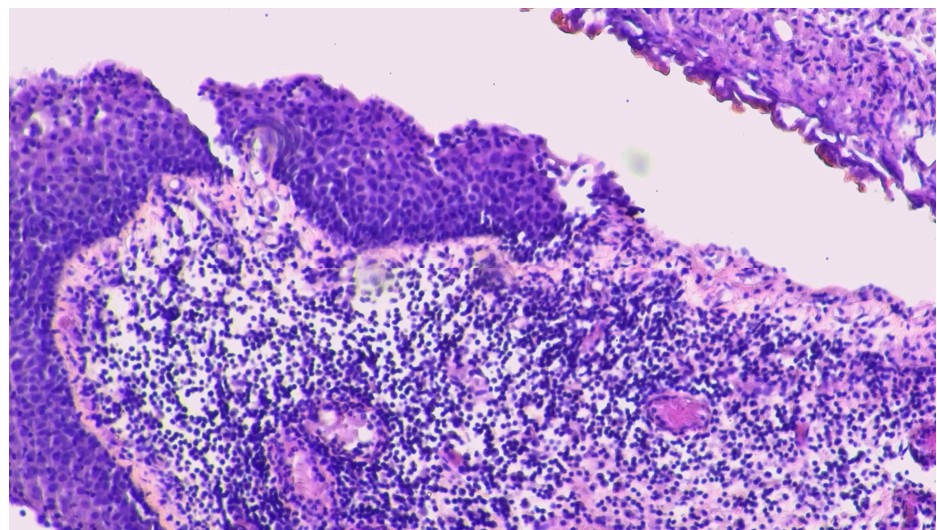

**Figure 4.** Bladder biopsy. Legend: biopsy evidence of inflammatory process.

Over the following months, regular monitoring with CEA (6.5 ng/mL), Ca19.9 (27 UI/mL), CT scans (with FDG-PET performed in the case of any suspicious CT scan), and brain MRI every 3–4 months were performed with a complete response 37 months after diagnosis (Figures 5 and 6). The last diagnostic imaging was performed in November 2024.

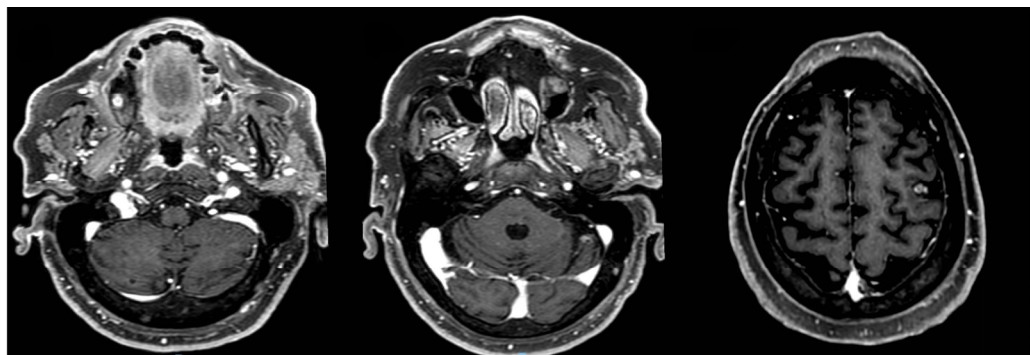

**Figure 5.** MRI response to treatment. Legend: brain MRI complete response to treatments for cerebellar and frontal lesions 35 months after diagnosis.

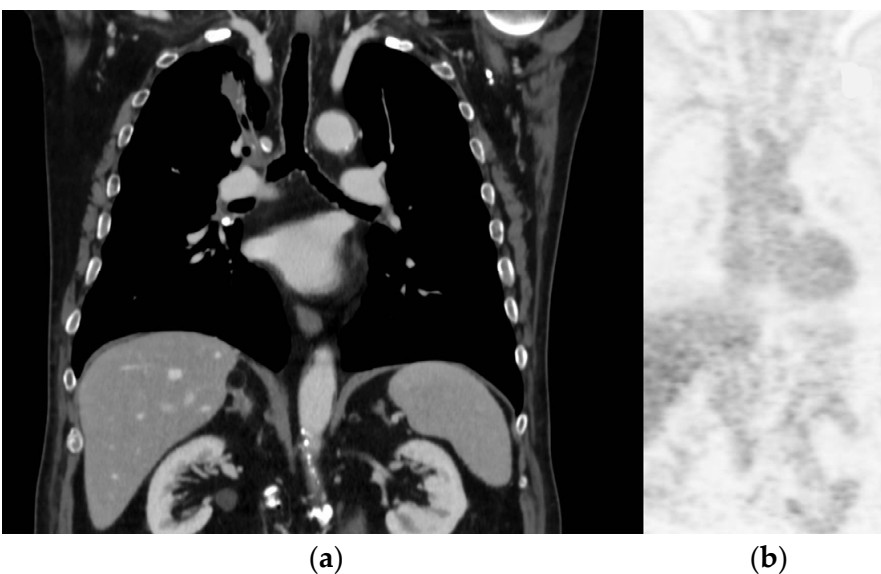

(**a**)  (**b**)

**Figure 6.** Response to treatment. Legend: (**a**) CT scan mediastinal and lung complete response to treatments 35 months after diagnosis; (**b**) PET-FDG complete response to treatments 35 months after diagnosis.

## 3. Discussion

The management of patients with oncogene-negative and PDL1-negative NSCLC poses a significant clinical challenge [4]. Furthermore, there is increasing evidence of an oligometastatic state in NSCLC in terms of association with local treatment and immunotherapy [5]. In addition, the role of PD-L1 expression is intricate and multifaceted [9–11]. It is important to note that cases where patients with metastases experience a complete response to treatment and remain disease-free for over two years are still uncommon and anecdotal in NSCLC [10–12]. Our patient's prognosis was poor due to the presence of synchronous brain metastasis (BM), with a median OS of less than six months in untreated individuals [13,14].

The impressive and long-lasting metabolic complete response seen in this patient with stage IV lung adenocarcinoma with BM, despite the absence of mutations (PDL1-, EGFR-, ROS-, and ALK-negative), deserves careful consideration. This case provides valuable insights into the changing landscape of LC treatment.

### 3.1. The Role of RT in Oligometastatic NSCLC

The role of ablation treatment, including surgery and RT, has been assessed in the current scenario in a systematic review [3]. The review evaluated 49 relevant studies involving 2176 patients. These studies focused on patients with oligometastases (with 1 to 5 lesions) who underwent surgical metastasectomy and stereotactic ablative radiotherapy (SBRT). Sixty percent of all the studies included patients with exclusive brain metastases. In 82% of patients, the primary tumor was controlled, with a median OS of 14.8 months for all patients (range 5.9–52) and of 19 months among patients with a controlled primary tumor [3].

The integration of SBRT for persistent lesions highlights the potential of multimodal treatment. It offers high-precision, localized treatment that could benefit specific lesions and increase overall survival. The SABR-COMET phase II trial included patients with various types of primary tumors, with stage IV NSCLC being one of the most common, accounting for 18% of the total. Ninety-nine patients with controlled primary tumors and five or fewer metastatic lesions were eligible. They were divided into two groups for comparison: 66 in the SABR-plus-SOC arm and 33 in the palliative-SOC arm. Long-term follow-up showed a remarkable 5-year OS rate of 42% (95% CI, 28–56) in the SABR-plus-SOC arm compared to 18% (95% CI, 6–34; $p = 0.006$) in the palliative-SOC arm [13].

Based on this analysis, the present case was treated by SBRT for all three BMs.

Radiation can have both suppressive and stimulating effects on the immune system. The stimulating effects, such as releasing tumor antigens when radiation kills tumor cells, may help trigger an immune response against the tumor throughout the body; however, these responses are rare. To improve this effect, some suggest combining radiation with immune checkpoint inhibitor therapy for stage IV NSCLC. These are areas of ongoing preclinical and clinical investigation [5,14].

Atezolizumab is an immune checkpoint inhibitor, a type of targeted therapy drug. It is a monoclonal antibody that works by binding to the protein PD-L1 on the surface of some cancer cells. This prevents cancer cells from suppressing the immune system, allowing the immune system to attack and kill the cells. Data regarding the BBB penetration of ICIs are limited [15,16]. The most compelling evidence supporting the effectiveness of immune checkpoint inhibitor (ICI) therapy for active BMs comes from a phase II trial of pembrolizumab conducted by Goldberg et al. This study involved 42 patients with advanced non-oncogene-driven NSCLC and untreated, asymptomatic BMs ranging in size from 5 to 20 mm. Patients with PD-L1 $\geq$ 1% had a 29.7% intracranial objective response rate (ORR), with seven patients showing partial response on subsequent imaging and four achieving complete response. Their median OS was 9.9 months (cohort 1). In contrast, responses in the brain metastases of patients with PD-L1 < 1% were either not present or could not be evaluated (cohort 2) [15].

### *3.2. The Role of Hypo-RT or Ablative RT in Anti-Tumor Immunity*

Emerging evidence suggests that hypo-RT may have a solid anti-tumor effect by causing tumor cell death through anti-tumor immunity and vascular damage [14]. RT not only directly impacts the irradiated area but also stimulates the immune response indirectly because of the low irradiation doses outside the target area. Several studies have shown that hypo-RT can boost the body's anti-tumor immune responses; this happens because hypo-RT causes tumor cell death, normalizes abnormal blood vessel growth, and releases tumor-associated antigens and pro-inflammatory cytokines.

However, some investigations in animal models have advised that the tumor microenvironment (TME) changes created by RT may develop an immunosuppressive TME, which could promote tumor invasion and spread in some cases.

After hypo-RT, tumor cells expose more specific molecules on their surfaces recognized by the immune system, with increased sensitivity to T-cell-mediated cell death; in addition, hypo-RT increases pro-inflammatory molecules and danger signals in the TME. Consequently, the CD8+ T and dendritic cells are triggered and recruited into the tumor; in this context, these immune cells play a key role in anti-cancer immune responses [17].

Moreover, for the present case, local hypo-RT was chosen.

### *3.3. The Volume Reduction in RT Increases the Immunity Response*

The lymph node controls the interactions and movement of various immune cells, including T, B, and antigen-presenting cells. RT may enhance the ability of cytotoxic T cells to kill tumors in the TME by possibly increasing the expression of primary histocompatibility complex class I in tumor cells. However, tumor-draining lymph nodes (TDLN) are crucial for RT-induced immune stimulation and abscopal effects. Phenotypically, dysfunction of the TDLN caused by surgery, irradiation, pharmacological inhibitors, or genetic ablation significantly reduces the anti-tumor effects of RT and immunotherapy [18]. Marciscano et al. [19] showed that irradiating elective TDLN reduced immune-RT infiltration by regulating chemokine signaling. This led to an unfavorable TME in the primary tumor.

Thus, the RT volume reduction (involved field RT) was utilized in the present case.

### *3.4. The Timing of ICI/RT*

The first demonstration of a clinical benefit for the combination treatment of ICI and RT was evidenced in a retrospective analysis of highly immunogenic cancer lesions (in

particular, metastatic melanoma, NSCLC, and renal cell carcinoma), which compared the efficacy of RT alone or RT in combination with ICI [20]. The evaluation of the optimal timing strategy was analyzed in a comprehensive meta-analysis of numerous clinical evidence that retrospectively suggested the superiority of concurrent ICI and RT treatment over sequential treatment.

In this case, SBRT was the first approach used to treat BM, and then hypo-RT was performed between the 10th and 11th atezolizumab cycles.

### 3.5. The Role of ICI Acute Toxicity

ICIs induce a high risk of developing immune-related adverse events (irAEs) [21,22]. Different reports indicate that the incidence of irAEs can range from 60% to 85%, depending on the use of mono- or combination immunotherapy, affecting mostly the skin, endocrine glands, gastrointestinal tract, lungs, and liver. Hypophysitis and thyroid dysfunctions, as in the present case, are the most common endocrine side effects of immune checkpoint inhibitors, while Type 1 diabetes mellitus and adrenal insufficiency are less common side effects. Most patients usually recover from pituitary–thyroid and pituitary–gonadal axes dysfunction, while improvement in the pituitary–adrenal axis has been rarely observed.

On the other hand, a limited amount of literature exists on the incidence, time of onset, and risk factors for multiorgan system irAEs, which occurred in about 5% of ICI-treated patients. It is still debated whether there is a direct correlation between ICI effectiveness and the degree of treatment-induced toxicity. Reports showed that irAEs are strongly correlated with better survival and higher response rates in patients with melanoma, advanced gastric cancer, or NSCLC receiving anti-PD-1 therapy.

A recent analysis was conducted using data from four clinical trials of atezolizumab in patients with NSCLC. The study found that 5.4% of patients experienced multiorgan irAEs, with "skin" or "laboratory" being the most common. However, it was also observed that patients with multiorgan irAEs had improved OS compared to those without irAEs (hazard ratio, 0.47; 95% CI, 0.28–0.78; $p < 0.0001$). There was no significant difference in progression-free survival between the two groups (hazard ratio, 0.92; 95% CI, 0.62–1.35; $p = 0.74$).

Our case reported multiorgan irAEs (psoriasis, hypothyroidism, and cystitis with ulcer-necrotic bladder), showing an extended response 19 months after the interruption of ICI.

### 4. Final Considerations

In the literature, no data regarding prolonged survival have been published for metastatic oncogene-negative, PDL1-negative NSCLC.

Patients with metastatic NSCLC have a short life expectancy despite the use of advanced medical and pharmaceutical technologies. In this situation, it is increasingly important to collect and study cases of patients with prolonged survival to examine their immunological characteristics. This will help search for improved treatment options for this group of patients.

### 5. Conclusions

In conclusion, this case report illustrates the potential role and efficacy of combining ICI with chemotherapy and the addition of RT in treating oncogene-negative, PDL1-negative NSCLC. Further research is needed to improve the outcomes of this population. It is essential to understand how local therapy, such as RT, affects tumors and metastases and the immune effects of hypofractionated or SBRT on the TME. Figuring out the best timing for combining immunotherapy and RT may improve treatment outcomes and help us discover new combination therapy strategies.

**Author Contributions:** Conceptualization, all authors; methodology, M.C., A.F., F.C.D.G. and V.D.; software, A.S. and A.F.; validation, I.B. (Isabella Bruno), M.P.C., R.C. (Roberta Carbonara), R.C. (Roberto Calbi), M.A.G., N.S., T.T., G.S., I.B. (Ilaria Bonaparte) and A.F.; formal analysis, A.F.; investigation, A.F. and G.S.; resources, M.C.; data curation, M.C. and A.F.; writing—original draft preparation, A.F.; writing—review and editing, A.S., R.C. (Roberta Carbonara), F.C.D.G. and V.D.; visualization, M.C. and G.M.L.; supervision, A.F., I.B. (Ilaria Bonaparte) and G.S. All authors have read and agreed to the published version of the manuscript.

**Funding:** This research received no external funding.

**Institutional Review Board Statement:** This study was conducted according to the guidelines of the Helsinki Declaration and approved by the Institutional Review Board of the General Regional Hospital F. Miulli (protocol code RT-AMB-2021).

**Informed Consent Statement:** Written informed consent has been obtained from the patient to publish this paper.

**Data Availability Statement:** Data requests can be directed to the corresponding author.

**Conflicts of Interest:** The authors declare no conflicts of interest.

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
