# Peer review of "Metabolic Complete Response of Metastatic Oncogene-Negative, PDL1-Negative Non-Small Cell Lung Cancer After Chemo-Immunotherapy and Radiotherapy: A Case Report"

_curroncol, doi:10.3390/curroncol31120598_

Round 1

Reviewer 1 Report

Comments and Suggestions for Authors

This is a rare report and I believe it would be worthy of publication if the following comments were addressed.

Line 72-73

For SRT 24Gy/3 fractions, is it Gamma Knife, Cyber Knife, or Linac? This is a rare case, so please provide a detailed description of the type of SRT.

Line 91

Please align the wording of the following parts with the other parts. From 29.8.2022 to 2.2.2023,

Line 230-249

The conclusion is too long. Please summarize the clinical significance and novelty of this report more concisely.

Do you think there was an abscopal effect in this case? Please mention this in your Discussion.

Author Response

Reviewer #1: This is a rare report, and I believe it would be worthy of publication if the following comments were addressed.

Line 72-73

For SRT 24Gy/3 fractions, is it Gamma Knife, Cyber Knife, or Linac? This is a rare case, so please provide a detailed description of the type of SRT.

Authors: Thanks a lot for this kind consideration of our manuscript. We improved the text as required by the referee.

We added these sentences:

  • “He started LINAC-based SRT.”
  • The gross tumor volume (GTV) was defined on contrast-enhanced T1-weighted sequence MRI and was assumed equal to the clinical target volume (CTV). The planning target volume (PTV) was obtained from the GTV plus an isotropic margin of 1 mm in all directions. The treatment was performed with 6X Flattening Filter Free (FFF) beam and Volumetric Modulated Arc Therapy (VMAT) with noncoplanar arcs, optimized for a True-Beam™ (Varian Medical System). HyperArcTM approach was used for the simultaneous treatment of the 2 cerebellar lesions [7]. Prescription dose, normalization and optimization were according to ICRU91 guidelines. During RT, IGRT with cone-beam CT (CBCT) and real-time surface-guided RT using AlignRT® was performed daily prior to and during the RT session [7].

Reviewer #1: Line 91

Please align the wording of the following parts with the other parts. From 29.8.2022 to 2.2.2023,

Authors: thanks to the referee, we rephrased as follows:

“From August 29th, 2022, to February 2nd, 2023,”

Reviewer #1: Line 230-249

The conclusion is too long. Please summarize the clinical significance and novelty of this     report more concisely.

Authors: we agree with the referee and for this reason, we modified the text as follows:

  1. Final considerations

In the literature no data with prolonged survival was published for metastatic oncogene-negative, PDL1-negative NSCLC.

Patients with metastatic NSCLC have a short life expectancy despite the use of advanced medical and pharmaceutical technologies. In this situation, it is increasingly important to collect and to study cases of patients with prolonged survival to examine their immunological characteristics. This will help search for improved treatment options for this group of patients.

  1. Conclusions

In conclusion, this case report illustrates the potential role and efficacy of combining ICI with chemotherapy and the addition of RT in treating oncogene-negative, PDL1-negative NSCLC. Further research is needed to improve the outcomes of this population. It's essential to understand how local therapy, such as RT, affects tumors and metastases and the immune effects of hypofractionated or SBRT on the TME. Figuring out the best timing for combining immunotherapy and RT may improve treatment outcomes and help us discover new combination therapy strategies.

Reviewer #1: Do you think there was an abscopal effect in this case? Please mention this in your Discussion

Authors: thanks to the referee for this interesting question.

Yes, that's likely the case. Initially, the patients had three brain metastases and one bone metastasis. After administering stereotactic body radiation therapy (SBRT) for the brain metastases, the MRI showed a reduction in both the brain and bone metastases, probably due to the abscopal effect. However, since the bone metastasis was the only one that was not irradiated, the statement that an abscopal effect may have occurred seems questionable in this context.

Reviewer 2 Report

Comments and Suggestions for Authors

Interesting case report, well presented and discussed.

Author Response

Thanks a lot for this consideration of our manuscript

Reviewer 3 Report

Comments and Suggestions for Authors

Abstract: the systemic therapy doesn’t correspond to the mentioned on the case report description. The designation of oncogene negative should be review, since the only genetic alterations analyzed was ALK, ROS1, EGFR

Introduction: good contextualization

Case report:

·       What treatment was done to bone lesion? was it irradiated?

·       What technique was used for genetic analysis of ALK, ROS1, EGFR?

·       Consider adding the response evaluation using RECIST criteria. – line 84 and 87

·       Line 87 – size of progression or % by RECIST criteria

·       Specify data of onset of adverse events: cystitis, cutaneous and hypothyroidism and respective treatments. The Cystitis was only in July 2024? More than 1 one after suspension of IO?

·       How was the follow-up after treatment suspension?  PET CT and MRI were performed with what frequency?

Discussion

·       Line 124 – no mention to EGFR

·       Discussion provides a good revision of literature and justification for the used strategy.

Conclusion

·       Consider rephrase the first sentence, because, since the introduction of new treatment options for oncogene and non-oncogene addicted metastatic NSCLC, like immunotherapy and target therapy, the overall survival has increased significantly. For that matter, the affirmation that OS rarely goes further than 24 months isn’t correct.

·       Line 231-235 – consider omitting this example. Brain metastases in EGFR positive patients present a distinct scenario, where target therapies are most of the time effective. The example of use of temozolomide is off-label and doesn´t fits well in this conclusion.

Author Response

Reviewer #3: Abstract: the systemic therapy doesn’t correspond to the mentioned on the case report description. The designation of oncogene negative should be review, since the only genetic alterations analyzed was ALK, ROS1, EGFR

Authors: thanks to the referee for the observation, we corrected the text in the abstract with the correct systemic therapy used:

“…with Atezolizumab.”

KRAS, RET, HER2, BRAF, PIK3CA, NRAS, MET and MEK were analyzed too and all were negative.

Reviewer #3: Introduction: good contextualization

Authors: Thanks a lot for this consideration of our manuscript.

Reviewer #3: Case report:

What treatment was done to bone lesion? was it irradiated?

Authors: thanks to the referee.

The bone lesion was not irradiated. The follow-up FDG-PET in March 2023 showed a complete response.

What technique was used for genetic analysis of ALK, ROS1, EGFR?

Authors: thanks to the referee for the question, we added the sentence in the text

EGFR genes and ALK/ROS1 mutations were detected using DNA sequencing techniques and the amplification-refractory mutation system. The mRNAs of ROS1 and ALK fusions were examined using polymerase chain reaction techniques and a fusion gene detection kit.

Consider adding the response evaluation using RECIST criteria. – line 84 and 87

Authors: thanks to the referee for the observation, we modified the text as follow:

The response can be defined as partial using the RECIST (Response Evaluation Criteria in Solid Tumors) criteria.

Line 87 – size of progression or % by RECIST criteria

In June 2022, imaging showed a growth in lung mass and mediastinal nodal size with PD defined by RECIST.

Specify data of onset of adverse events: cystitis, cutaneous and hypothyroidism and respective treatments. The Cystitis was only in July 2024? More than 1 one after suspension of IO?

Authors: thanks to the referee, we modified the text as follow:

The patient developed hypothyroidism (TSH 64 UI/ml, range 0.27-4.2) with heart failure treated with levothyroxine, Grade 4 dermatitis from psoriasis, and pruritus unresponsive to antihistamines, which was managed with glucocorticoids. The patient resolves the side effects 3-4 months after the Atezolizumab interruption. Moreover, during months from March 2023 on, he reported urinary symptoms, including mild hematuria and cystitis. In July 2024, patients reported chronic cystitis with G3 hematuria and underwent a cystoscopy, and a biopsy was performed.

How was the follow-up after treatment suspension?  PET CT and MRI were performed with what frequency?

Authors: thanks to the referee, we added data on follow-up as follows:

Over the following months, regular monitoring with CEA (6.5 ng/ml), Ca19.9 (27UI/mL), CT scan (FDG-PET was performed in case of any suspicious at CT scan) and brain MRI every 3-4 months were performed with a complete response 37 months after diagnosis.

Reviewer #3: Discussion

  • Line 124 – no mention to EGFR

Authors: thanks to the referee, we improved the manuscript by adding EGFR in text.

“… (PDL1, EGFR, ROS, ALK-negative). “

Reviewer #3: Discussion provides a good revision of literature and justification for the used strategy.

Authors: Thanks a lot for this consideration of our manuscript.

Reviewer #3: Conclusion

Consider rephrase the first sentence, because, since the introduction of new treatment options for oncogene and non-oncogene addicted metastatic NSCLC, like immunotherapy and target therapy, the overall survival has increased significantly. For that matter, the affirmation that OS rarely goes further than 24 months isn’t correct.

Line 231-235 – consider omitting this example. Brain metastases in EGFR positive patients present a distinct scenario, where target therapies are most of the time effective. The example of use of temozolomide is off-label and doesn´t fits well in this conclusion.

Authors: reading the text, we agree with the referee. For this reason, we modify the text as follows:

  1. Final considerations

In the literature no data with prolonged survival was published for metastatic oncogene-negative, PDL1-negative NSCLC.

Patients with metastatic NSCLC have a short life expectancy despite the use of advanced medical and pharmaceutical technologies. In this situation, it is increasingly important to collect and to study cases of patients with prolonged survival to examine their immunological characteristics. This will help search for improved treatment options for this group of patients.

  1. Conclusions

In conclusion, this case report illustrates the potential role and efficacy of combining ICI with chemotherapy and the addition of RT in treating oncogene-negative, PDL1-negative NSCLC. Further research is needed to improve the outcomes of this population. It's essential to understand how local therapy, such as RT, affects tumors and metastases and the immune effects of hypofractionated or SBRT on the TME. Figuring out the best timing for combining immunotherapy and RT may improve treatment outcomes and help us discover new combination therapy strategies.

Round 2

Reviewer 3 Report

Comments and Suggestions for Authors

the corrections and information added have improved the article